# Adaptive Leader-Follower Formation Control of Under-actuated Surface Vessels with Model Uncertainties and Input Constraints

**Alireza Riahifard [1], Seyyed Mohammad Hosseini Rostami [2] , Jin Wang [3,4] and Hye-Jin Kim [5,***

[1] Department of Electrical and Computer Engineering, Science and Research Branch, Islamic Azad University, Tehran 1477893855, Iran; alireza.riahifard@srbiau.ac.ir
[2] Department of Electrical and Computer Engineering, Shiraz University of Technology, Shiraz 7155713876, Iran; seyyed.mazandaran@gmail.com
[3] Hunan Provincial Key Laboratory of Intelligent Processing of Big Data on Transportation, School of Computer & Communication Engineering, Changsha University of Science & Technology, Changsha 410004, China; jinwang@csust.edu.cn
[4] School of Information Science and Engineering, Fujian University of Technology, Fuzhou 350118, Fujian, China
[5] Business Administration Research Institute, Sungshin W. University, Seoul 02844, Korea
* Correspondence: hye-jinkim@hotmail.com; Tel.: +82-022-2870-097

**Abstract:** This paper deals with the leader-follower formation control of underactuated autonomous surface vehicles in the presence of model uncertainties and input constraints. In a leader-follower formation, an autonomous surface vehicle (ASV) called leader tracks a pre-described trajectory and other ASVs called followers that are controlled to follow the leader with a desired distance and desired relative bearing. To this end, some adaptive robust techniques are adopted to guarantee the robustness of the closed-loop system against model uncertainties, external disturbances, and input saturation constraints. Based on the Lyapunov synthesis, it is proven that with the developed formation controllers, the closed-loop system is stable and all the formation errors converge to a small neighborhood of zero. Simulation results demonstrate the effectiveness of the proposed method.

**Keywords:** autonomous surface vehicles; adaptive robust control; model uncertainty; leader-follower formation

## 1. Introduction

### 1.1. Motivation

The control and modeling of batches or networks of multiple vessels (air and sea) is a new topic that has received much attention in recent years from the community of control researchers. The idea under discussion is that a group of vessels in a given category or network, performance and capability are much better than moving alone on any of the vessels and can perform more tasks: autonomous surface vessels are robotic boats or ships that respond to environmental changes and perform various tasks with minimal manpower intervention. Like many advanced systems capable of civilian use, the development of autonomous surface vessels for military applications has also begun.

Control of multiple autonomous surface vehicles (ASVs) to operate together as a team was first received attention of control engineers in 1991 [1]. Since the early 1990s, ASVs attracted great attention in system and control. Autonomous surface vessels are robotic ships that can react to environmental changes and fulfill different tasks with minimal human intervention. Individual

surface vehicles that are connected as a network with a specific shape, can perform various tasks. Relevant applications include automatic ocean exploration, environmental monitoring, disaster search and rescue, surveillance of territorial waters, underway ship replenishment, and so on. In general, formation control is the coordination of a group of robots to create and maintain a seal arrangement with the specified format [2–5]. To achieve the desired formation between UAVs, several methods have been proposed. Among these control schemes, the leader-follower strategy seems to be much preferred in practice due to its simplicity and scalability and has been studied by many researchers [6,7]. In a formation with leader-follower configuration, one or more ASVs is selected as leaders, which are responsible for guiding the formation, and the rest of the ASVs are controlled to follow the leaders. The control objective is to make the follower ASVs track the leaders with some prescribed offsets. In recent years, many visual object tracking techniques based on correlation filters [8,9] and deep learning [10,11] have emerged. Other advantages of the leader-follower formation are simplicity, understandability and its easy implementation.

A sample formation is shown in Figure 1. As can be seen, one leader (red) and four followers (blue and green) arranged in a particular formation to form a triangular motion without conflict to each other and with predetermined moving.

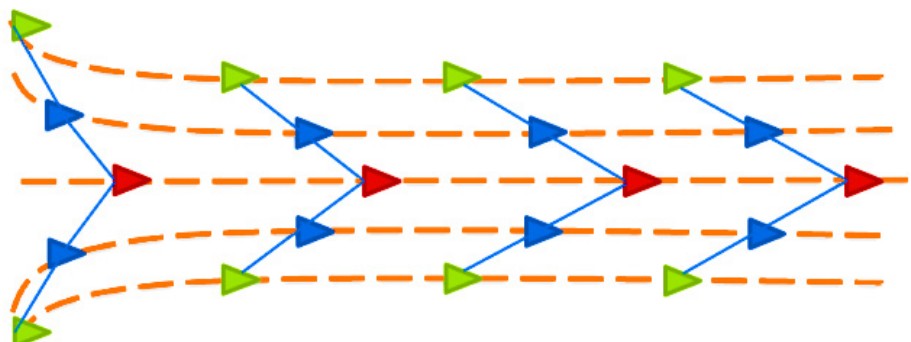

**Figure 1.** Leader-follower formation with triangle pattern in two dimensions.

## 1.2. Related Works

Over the past decade, from 2002 to 2010, most studies on formation control of ASVs are within the framework of leader-follower strategy [12,13]. Nonlinear formation control of surface vehicles was first considered on paper [12] that used backstopping to keep formation. In 2003 and 2004, some works presented for underactuated surface vessel (i.e., the surface vessels with three degrees of freedom) so that the proposed controllers were adjustable with realistic models of vessels [14,15]. The formation control problem with and without communication constraints and also considering the time delay for marine vessels and submarines with full actuated [16,17] and underactuated [16–18] has been studied from 2003 to 2007. Sliding mode techniques for controlling leader-follower underactuated surface vehicles have been considered in [19]. By using the theory of pioneers and parallel integrals [20], a directed leader-follower controller for full actuated surface vessels has been proposed. In practice, most of the existing control methods that have been presented for dynamical systems, are not implementable. A considered model may not be accurate and a system dynamic may be exposed to bounded disturbances or unmolded dynamics. In addition, external disturbances always affect the behavior of a dynamic system. Based on the distance techniques in formation control, adaptive controllers are proposed to keep formation among vessels [21,22]. Using neural networks, an adaptive controller is designed for the first time for surface vehicles to deal with unknown dynamics in formation control in [23,24]; however, the results obtained for full actuated surface vessels were not appropriate for underactuated surface vehicles. Formation control of ASVs with unknown dynamics of leader and local dynamic of followers has been considered in [25]. However, the controller design was very complicated due to the numerical calculating of partial derivatives of control signals. In addition of what was studied in

previous papers, the control inputs in surface vessels (e.g., torques that apply to move a vessel) cannot adopt any value. The torque that is applied by a motor on blades and arms of a vessel is exposed to actuator saturation. In the other words, input constraints are not considered in previous works and as a result, most of controllers which have been designed in previous studies for trajectory tracking or to keep formation, are not implementable to real surface vessels. In this paper, the problem of leader-follower formation control for a group of ASVs has been considered. To this end and after modeling of leader-follower formation, first considering uncertainty, disturbances, and unmolded dynamics in the formation model, an adaptive robust controller is designed to keep formation. Then, this controller is modified to keep formation in the presence of input saturation constraints in addition of model uncertainties. It is related to the maneuver multiple autonomous surface vehicles that guided by a virtual leader moving in a parametric path in [26].

This paper addressed the maneuver of the various distributions of autonomous surface vehicles (ASVs) that currently have an unknown and state-restricted movement of motion. A design method for maneuvering the distribution of multiple ASVs based on neurodynamic optimization and fuzzy approximation is presented. The neuro-optimization method is able to obtain an optimal guidance signal for proper speed constraints and reducing control effort. A predictor is designed so that a fuzzy system is used to approximate unknown kinetic energy based on input and output data. The Stability of the closed-loop system has used the virtue of cascade theory. The continuous path problem using the error-constrained line-of-sight (ECLOS) guidance method is discussed in [27]. ECLOS guidance is applied to a Surface Vessel to control the path under the uncertainty, actuator saturation, and faults. It is shown that with the proposed method, the requirement for an error limit is never violated for all time. By using a non-linear disturbance observer, all uncertainties are estimated and compensated. The proposed track control can ensure that closed-loop system state is ultimately limited and final range can be arbitrarily small by arbitrary selection of parameters. The issue of controlling the maneuvering of sea surface vehicles by numerous virtual leaders, stating that movement is guided along with various path parameters, is examined in [28]. In addition, only a handful of sequential vehicles have access to the information of its virtual leaders.

The problem of controlling the formation of a limited follower leader for a class of independent surface vessels with a range of line of sight (LOS) and angle constraints are considered in [29]. A new control structure ensures convergence into small areas arbitrarily in the zero range for a limited time for tracking errors, while the required constraints on the LOS range and angle are not violated. In the paper [30], the tracking path of the controller problem of a fully actuated surface vessel is limited to the asymmetric input and the output is addressed. An adaptive control approach is used barrier lyapunov function (BLF) and nussbaum function to implement trajectory tracking for a fully actuated surface vessel point under the asymmetric constraints of input and output. To adaptive an asymmetric saturation function, almost a tangent function is used in the form of smooth hyperbolic. The command system and auxiliary design system were adopted to avoid complex calculation, derived from virtual control and ensure its limitations. With the proposed approach, proved that tracking errors of the closed-loop system are ultimately limited.

A controller for tracking path for a fully actuated surface vessel in the presence of output constraints and uncertainty using adaptive asymmetric barrier Lyapunov function (ABLF) and neural networks (NNs) [31–33]. This method proves that in the suggested proposal law, closed-loop system signals, threshold tracking are obtained, and output constraints are not violated too much.

This paper [33] provides a way to exploit sea surface vessels with predicted tracking controls. By introducing an error function, the limited tracking control of the main vessel turned into the stability of an unrestricted system. Adaptive stability of the neural networks (NN) tracking control is designed for unspecified sea surface vessels. The proposed controller ensures that the output tracking errors of the system have transient preset and stable state control. In persistent excitation (PE) conditions, a NN adaptive controller is able to gain knowledge of the understanding, expression, and storage of the unknown system dynamics in the stable state control process. The neural controller was created using

the stored knowledge, without the need to read the controller parameters, to achieve improved system control function. Controller for Ocean Surface Vessels is designed in [34] with accurate information about external impairments that operate on systems, that should be accurately estimated. To address this challenge, a new estimation approach was presented. The non-linear prediction rule was originally proposed. Under this rule, a sliding mode method is created based on the developed observer. The estimation error ensures to be stable in a limited time. Real disturbance can be estimated precisely with the error of zero estimate after a limited time. A quick and accurate estimate of the disruption is guaranteed. When vessels dynamic is also affected by the system's uncertainty, the proposed approach is able to estimate the total value of the uncertainty and disruptions simultaneously. An adaptive control for trajectory tracking autonomous underwater vehicle (AUV) using neural networks (NNs) in the discrete-time domain in [35]. A NN-based reinforcement learning algorithm is used to solve an unknown disorder, uncertainty, and non-linear input control. A leader-follower method with an asymmetric limitation in the range and tolerance of the LOS angle is presented to track error between leader and follower for sub-active surface vessels is presented in [36]. It is assumed that the information speed of the leader is not available for tracking purposes. For this purpose, a reconstruction module is designed to accurately inject this information into a controller as a reference for follow-up by the vehicle follower. The control of formation for Nonholonomic mobile robots with heterogeneous uncertainties is studied in [37]. The formation mechanism is based on the leader-follower plan. The integral sliding mode control (ISMC) method is used to avoid uncertainties incompatible when robots are formed. The theoretical analysis, using the Lyapunov method, proves that the sliding surface vector of the coordinated formation control system with anomalous uncertainty is locally asymptotic stable. Conditions that are accessible by SMC are guaranteed using the ISMC methodology. The control method provided for the maneuvers of forming a multi-robot system consisting of three robots in two predicted paths in the presence of asymmetric uncertainty. Simulation results show the effectiveness, feasibility, and robustness of the ISMC method. Other robust methods have also been used for uncertainty removal [38–40]. Energy replenish method for robots is provided in [41,42].

The authors proposed a supervised learning method to do the classification that is very interesting and can be applied to our method [43,44]. The authors in [45–53] gave a novel mobile sink-based method to optimize system performance, which can be referred to.

This paper is organized as follows: Kinematic and dynamic model of autonomous surface vehicles are given in Section 2. Moreover, the leader-follower formation model of ASVs obtains in this section. Adaptive robust formation control that is the main result of this paper is designed for the considered leader-follower model in Section 3. Simulation results are given in Section 4 to validate the proposed methodology. Finally, the paper is concluded in Section 5.

## 2. Problem Formulation

In this section, first, the dynamic and kinematic models of surface vessels are expressed. Then, the leader-follower formation model of ASVs is obtained. This model is used for designing the controllers in the next sections.

### 2.1. Kinematic Model of Surface Vessels

An exact model of surface vehicles is with six degrees of freedom, where three parameters indicate the position of the center of mass of the vessel and the other three parameters determine the orientation of the body coordinate frame. However, the model of surface vehicles has been considered as underactuated in this paper in which, the number of controlled degree of freedoms are more than of independent input controls. Three variables that we work on it in this article is shown in Figure 2.

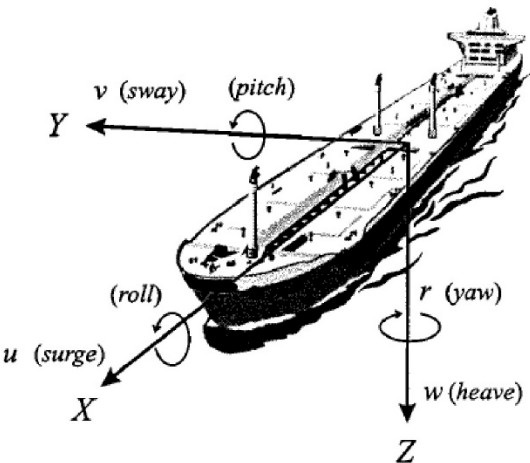

**Figure 2.** A representation of the variables of an autonomous surface vehicle (ASV).

Consider a group of number N ASVs. The kinematic model for $i$-th surface vehicle is described in [25]:

$$\dot{x}_i = u_i \cos \varphi_i - v_i \sin \varphi_i$$
$$\dot{y}_i = u_i \sin \varphi_i + v_i \cos \varphi_i \qquad (1)$$
$$\dot{\varphi}_i = r_i$$

where $x_i$ and $y_i$ indicate position and $\varphi_i$ indicates the orientation of the $i$-th surface vehicle in the earth fixed frame. $u_i$, $v_i$, and $r_i$ respectively surge, sway, and heave velocities. Considering the state vector as $\eta_i = [x_i, y_i, \varphi_i]^T \in \mathcal{R}^3$ and the velocity vector as $V_i = [u_i, v_i, r_i]^T$, the model in Equation (1) can be rewritten as follow:

$$\dot{\eta}_i = S_i(\eta_i) V_i(t), (i = 1, 2, \ldots, N) \qquad (2)$$

where

$$S_i(\eta_i) = \begin{bmatrix} \cos \varphi_i & -\sin \varphi_i & 0 \\ \sin \varphi_i & \cos \varphi_i & 0 \\ 0 & 0 & 1 \end{bmatrix}. \qquad (3)$$

### 2.2. Dynamic Model of Surface Vessels

The dynamic model of each surface vehicle in the direction of its movement in a group of N ASVs, described by [30]:

$$M_i \dot{V}_i + C_i(V_i) V_i + D_i V_i + \tau_{iw}(t) = \tau_{ai}(t) \qquad (4)$$

where $\tau_{ai}(t) = [\tau_u, 0, \tau_r]^T$, $\tau_{iw}(t)$ is an external disturbance and the matrices of the model are as follows:

$$M_i = \begin{bmatrix} m_{11} & 0 & 0 \\ 0 & m_{22} & 0 \\ 0 & 0 & m_{33} \end{bmatrix}, D_i = \begin{bmatrix} d_{11} & 0 & 0 \\ 0 & d_{22} & 0 \\ 0 & 0 & d_{33} \end{bmatrix}, C_i = \begin{bmatrix} 0 & 0 & -m_{22}v_i \\ 0 & 0 & m_{11}u_i \\ m_{22}v_i & -m_{11}u_i & 0 \end{bmatrix} \qquad (5)$$

It is worth mentioning that the sway velocity of ASVs is bounded as: $\text{Sup}_{t \geq 0} \|v_i\| < B_v$, where $B_v$ is a positive bounded constant. Therefore, in the velocity vector $V_i = [u_i, v_i, r_i]^T$, only the surge and heave velocities, i.e., $u_i$ and $r_i$ are important to be controlled and velocity in the direction of $v_i$ remains bounded due to the existence of a friction coefficient [51]. Define a new velocity vector as below that only consists the surge and heave velocities (Figure 3).

$$v_i(t) = [u_i, r_i]^T \qquad (6)$$

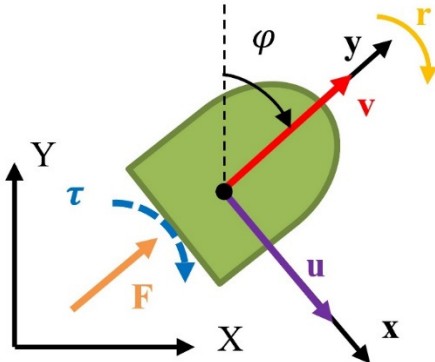

**Figure 3.** Dynamic model of an ASV with three degrees of freedom.

Therefore, considering the new torque vector as $\tau_{ai}(t) =, [\tau_u, \tau_r]^T$, the dynamic model of the ASV can be written as follows:

$$M_{1i}\dot{v}_i + C_{1i}(v_i)v_i + D_{1i}v_i + \tau_{w1\,i}(t) = \tau_{a\,i}(t) \tag{7}$$

where we have as follows:

$$M_{1i} = \begin{bmatrix} m_{11i} & 0 \\ 0 & m_{33i} \end{bmatrix} \tag{8}$$

$$C_{1i}(v_i) = \begin{bmatrix} 0 & -m_{22i}v_i \\ (m_{22i} - m_{11i})v_i & 0 \end{bmatrix} \tag{9}$$

$$D_{1i} = \begin{bmatrix} d_{11i} & 0 \\ 0 & d_{33i} \end{bmatrix} \tag{10}$$

In Equation (7), $\tau_{w1i}(t) \in \mathcal{R}^2$ denotes the vector of instant disturbances that caused by the environment (sea or ocean).

$$\tau_{w1i}(t) = \begin{bmatrix} \tau_{wu\,i}(t) \\ \tau_{wr\,i}(t) \end{bmatrix} \tag{11}$$

This vector is bounded as $\left\|\tau_{w1i}(t)\right\| \le \lambda_{\mathrm{r1i}}$ where $\lambda_{\mathrm{r1i}}$ is a positive constant.

Further, $\tau_{ai}(t) = [\tau_{ui}(t), \tau_{ri}(t)]^T$ denotes the input vector that is exposed to saturation constraint:

$$\left|\tau_{ai}(t)\right| \le \tau_{ai,max} \tag{12}$$

Property 1. The following properties can be proven for model (7).

$$x_i^T\left(\dot{M}_{1i} - 2C_{1i}(v_i)\right)x_i = 0 \,, \forall x_i, v_i \in \mathcal{R}^2 \tag{13}$$

$$\dot{M}_{1\,i} = C_{1\,i}(v_i) + C_{1i}^T(v_i) \,, \forall v_i \in \mathcal{R}^2 \tag{14}$$

$$C_{1i}(x_{1i})x_{2i} = C_{1i}(x_{2i})x_{1i}, \forall x_{1i}, x_{2i} \in \mathcal{R}^2 \tag{15}$$

$$C_{1i}(x_{1i} + x_{2i})y_i = C_{1i}(x_{1i})y_i + C_{1i}(x_{2i})y_i, \forall x_{1i}, x_{2i}, y_i \in \mathcal{R}^2 \tag{16}$$

$$\left\|C_{1i}(x_{1i})x_{2i}\right\| \le \lambda_{\mathrm{c1i}}\|x_{1i}\|\|x_{2i}\| \,, \forall x_{1i}, x_{2i} \in \mathcal{R}^2 \tag{17}$$

where $\lambda_{\mathrm{c1i}} \ge 0$ is a constant number.

### 2.3. Leader-Follower Formation Model

In the leader-follower formation model, the relative configuration between the leader and the $i$-th ASV follower is defined by $q_i = \begin{bmatrix} \rho_i(t) & \psi_i(t) \end{bmatrix}^T$, where $\rho_i(t)$ and $\psi_i(t)$ denote the relative distance and relative bearing respectively. To design this formation, position and relative bearing errors in the Earth-fixed frame $\{O_B, X_B, Y_E\}$ are transformed to the Body-fixed frame of $i$-th ASV, as follows (Figure 4).

$$e_i(t) = \begin{bmatrix} e_{i1}(t) \\ e_{i2}(t) \\ e_{i3}(t) \end{bmatrix} = \begin{bmatrix} \cos\varphi_i & \sin\varphi_i & 0 \\ -\sin\varphi_i & \cos\varphi_i & 0 \\ 0 & 0 & 1 \end{bmatrix} \begin{bmatrix} x_d - x_i \\ y_d - y_i \\ \varphi_d - \varphi_i \end{bmatrix}, i = 1, 2, \ldots, N \tag{18}$$

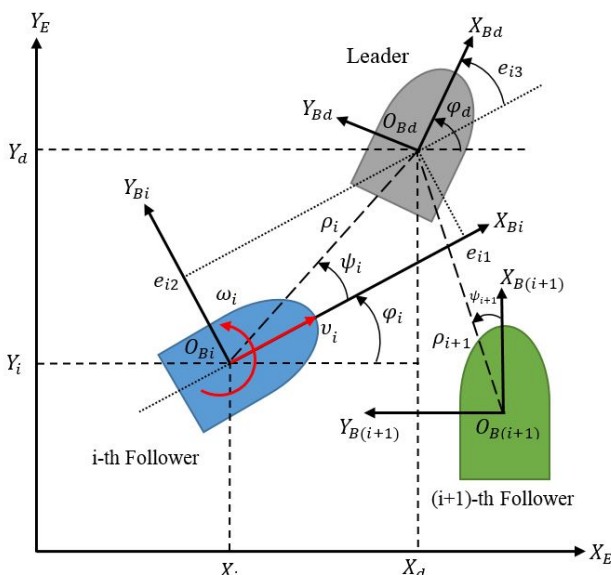

**Figure 4.** Leader-follower formation model.

Therefore, the distance and bearing of $i$-th ASV can be described as below:

$$q_i(t) = h_i(e_i(t)) = \begin{pmatrix} \rho_i(t) \\ \psi_i(t) \end{pmatrix} = \begin{pmatrix} \sqrt{e_{i1}^2(t) + e_{i2}^2(t)} \\ atan2(e_{i2}(t), e_{i1}(t)) \end{pmatrix}, i = 1, 2, \ldots, N \tag{19}$$

Time derivative of (19) results in (20) and (21) equations.

$$\dot{q}_i(t) = R_i v_i + \delta_i \tag{20}$$

$$\ddot{q}_i(t) = \dot{R}_i v_i + R_i \dot{v}_i + \dot{\delta}_i \tag{21}$$

where:

$$R_i = \begin{bmatrix} \dfrac{-e_{i1}(t)}{\rho_i(t)} & 0 \\ \dfrac{e_{i2}(t)}{\rho_i^2(t)} & -1 \end{bmatrix} \tag{22}$$

$$\delta_i = \begin{bmatrix} \dfrac{-v_i e_{i2} + u_d[e_{i1}\cos e_{i2} + e_{i2}\sin e_{i2}] + v_d[e_{i1}\sin e_{i2} + e_{i2}\cos e_{i2}]}{\rho_i(t)} \\ \dfrac{-v_i e_{i1} + u_d[e_{i1}\sin e_{i2} - e_{i2}\cos e_{i2}] + v_d[e_{i1}\cos e_{i2} - e_{i2}\sin e_{i2}]}{\rho_i^2(t)} \end{bmatrix} \tag{23}$$

From Equation (20) and (21), one can easily find $v_i$ and $\dot{v}_i$ as follows:

$$v_i = R_i^{-1}\left(\dot{q}_i - \delta_i\right) \tag{24}$$

$$\dot{v}_i = R_i^{-1}\ddot{q}_i - R_i^{-1}\dot{R}_i R_i^{-1}\left(\dot{q}_i - \delta_i\right) - R_i^{-1}\dot{\delta}_i \tag{25}$$

Substituting (24) and (25) into a dynamic model (7) results in the following model as the leader-follower formation model in ASVs.

$$M_i(e_i)\ddot{q}_i + C_i\left(e_i, \dot{q}_i\right)\dot{q}_i + D_i\left(e_i, \dot{u}_d, \dot{v}_d, \dot{v}_i\right) + R_i^{-T}\tau_{w1i}(t) = \overline{\tau}_{\text{ai}}(t) \tag{26}$$

where:

$$M_i(e_i) = R_i^{-T}M_{1i}R_i^{-1} \tag{27}$$

$$C_i\left(e_i, \dot{q}_i\right) = -R_i^{-T}M_{1i}R_i^{-1}\dot{R}_i R_i^{-1} \tag{28}$$

$$\overline{\tau}_{ai}(t) = R_i^{-T}\tau_{ai} \tag{29}$$

$$
\begin{aligned}
D_i\left(e_i, \dot{u}_d, \dot{v}_d, v_i\right) = {} & -R_i^{-T}M_{1i}\frac{\partial\left(R_i^{-1}\delta_i\right)}{\partial e_i}J_i R_i^{-1}\dot{q}_i + R_i^{-T}M_{1i}\frac{\partial\left(R_i^{-1}\delta_i\right)}{\partial e_i}J_i R_i^{-1}\delta_i \\
& -R_i^{-T}M_{1i}\frac{\partial\left(R_i^{-1}\delta_i\right)}{\partial e_i}\Delta_i - R_i^{-T}M_{1i}\frac{\partial\left(R_i^{-1}\delta_i\right)}{\partial u_d}\dot{u}_d - R_i^{-T}M_{1i}\frac{\partial\left(R_i^{-1}\delta_i\right)}{\partial v_d}\dot{v}_d \\
& -R_i^{-T}M_{1i}\frac{\partial\left(R_i^{-1}\delta_i\right)}{\partial v_i}\dot{v}_i + R_i^{-T}C_{1i}R_i^{-1}\dot{q}_i - R_i^{-T}C_{1i}R_i^{-1}\delta_i + R_i^{-T}D_{1i}R_i^{-1}\dot{q}_i - R_i^{-T}D_{1i}R_i^{-1}\delta_i
\end{aligned} \tag{30}
$$

## 3. Designing Adaptive-Robust Controller

In this section, first considering uncertainty, disturbances, and unmolded dynamics in the formation model, an adaptive robust controller is designed to keep leader-follower formation. In this controller, unknown parameters of formation model are estimated by an adaptive rule. In the second part of this section, this controller is modified to keep formation in the presence of input saturation constraints in addition to model uncertainties.

*3.1. Designing Adaptive-Robust Controller in the Presence of Model Uncertainties*

Considering $T_i = D_i\left(e_i, u_d, \dot{v}_d, \dot{v}_i\right) + R_i^{-T}\tau_{w1i}(t)$ as uncertainty, the model in (26) can be written as follows:

$$M_i(e_i)\ddot{q}_i + C_i\left(e_i, \dot{q}_i\right)\dot{q}_i + T_i = \overline{\tau}_{\text{ai}}(t) \tag{31}$$

Now, define $e_i = q_{di} - q_i$ and then $r_i = e_i + \dot{e}_i$ where $q_{di}$ indicates the desired formation for *i*-th ASV. We propose the following adaptive-robust controller for each of ASVs:

$$\overline{\tau}_{i\,a} = K_{iv}r_i + V_{iR} \tag{32}$$

where $K_{iv}$ is a diagonal positive definite matrix and $V_{iR}$ is obtained as follows:

$$V_{iR} = \frac{r_i\,\hat{\rho}_i^2}{\hat{\rho}_i\|r_i\| + \varepsilon_i} \tag{33}$$

where $\varepsilon_i$ and $\hat{\rho}_i$ are obtained from the following equations:

$$\dot{\varepsilon}_i = -K_{\varepsilon i}\varepsilon_i,\ \varepsilon_i(0) = 1,\ K_{\varepsilon i} \in \mathcal{R}^+ \tag{34}$$

$$\hat{\rho}_i = s_i\,\hat{\theta}_i \tag{35}$$

$$s_i = \begin{pmatrix} 1 & \|e_i\| & \|e_i\|^2 \end{pmatrix} \tag{36}$$

$$\hat{\theta}_i = \begin{pmatrix} \hat{\delta}_0 & \hat{\delta}_1 & \hat{\delta}_2 \end{pmatrix}^T \tag{37}$$

In these equations, $s_i$ is the regressor matrix and $\hat{\theta}_i$ denotes the estimated parameters vector that is obtained by the following adaptation rule.

$$\dot{\hat{\theta}}_i = \gamma_i s_i^T r_i \tag{38}$$

where $\gamma_i$ is an arbitrary positive constant value.

**Theorem 1.** *Consider the leader-follower formation model of ASVs in Equation (26) by applying the adaptive-robust controller in (32), the formation errors converge to zero, and estimated parameters in Equation (38) remain bounded.*

**Proof.** The proof is similar to the proof of applying a similar controller that has been proposed in [32] for solving the desired path following problem in manipulators. To prevent recurrence, the proof is omitted here. □

*3.2. Designing Adaptive-Robust Controller in the Presence of Model Uncertainties and Input Saturation Constraint*

First, consider the model in (26) that is based on the system states and rewrite it in the terms of errors. To this, substituting the error and its derivative that are, respectively, $e_i = q_{di} - q_i$ and $\dot{e}_i = \dot{q}_{di} - \dot{q}_i$ into (26) results.

$$M_i(e_i)\ddot{e}_i + C_i(e_i, \dot{e}_i)\dot{e}_i + \xi_i = -\overline{\tau}_{ai}(t) \tag{39}$$

where:

$$\xi_i = -M_i(e_i)\ddot{q}_d - C_i(e_i, \dot{e}_i)\dot{q}_d - D_i(e_i, \dot{u}_d, \dot{v}_d, \dot{v}_i) - R_i^{-T}\tau_{w1\,i}(t) \tag{40}$$

The aim is to design $\overline{\tau}_{ai}(t)$ in Equation (39) in such a way that $q_i \to q_d$ while $\overline{\tau}_{ai}(t)$ is bounded and the dynamic of the model is unknown. To this end, we use function *tanh* that is a bounded and continuous function. To reach the desired formation considering input constraint and unknown dynamic in model, we propose the following controller [52].

First, suppose that $\xi_i$ in Equation (40) is bounded as below:

$$\|\xi_i\| \le G_i \theta_i \tag{41}$$

where we have as follows:

$$G_i = \left[ \|\ddot{q}_d\|, \|\dot{q}_d\|\|\dot{q}_i\|, \|\dot{q}_i\|, \|R_i^{-T}(e_i)\| \right] \in R^{1 \times 4} \tag{42}$$

Here, $\theta_i$ is the parameters vector that includes mass, friction coefficient, and disturbances. In fact, $G_i \theta_i$ is the regressor form of the equation in (40). Now, define the following filtered error signal.

$$r_i = \dot{e}_i + \Lambda_{pi}\tanh(e_i) \tag{43}$$

where $\Lambda_p$ is a symmetric positive definite matrix. Finally, the bounded controller to keep formation is proposed as follows for the *i*-th ASV:

$$\overline{\tau}_{i\,a} = -K_{pi}\tanh(e_i) - K_{iv}\tanh(r_i) + u_{Ri} \tag{44}$$

where $k_{pi} \geq 1$, $K_{pi} = k_{pi}I_2$ and $K_{iv} \in R^{2\times 1}$ is a positive definite matrix. In addition, $u_{Ri}$ is obtained as follows:

$$u_{Ri} = -G_i\,\hat{\theta}_i \tanh\left(\frac{G_i\hat{\theta}_i r_i}{\gamma_d}\right). \tag{45}$$

The parameters $\hat{\theta}_i \in R^{4\times 4}$ in (45) are estimated by the following adaptation rule.

$$\dot{\hat{\theta}}_i = \Gamma_\theta G_i^T r_i - \Gamma_\theta \delta_\theta\left(\hat{\theta}_i - \theta_0\right) \tag{46}$$

where $\Gamma_\theta \in R^{4\times 4}$ is the controller gain, $\delta_\theta \in R^+$ and $\theta_0 \in R^{4\times 1}$ are the initial estimation of the system parameters. To prove the stability of the closed-loop error dynamics, will require the following features:

Property 2. The following properties can be proven for model in (39).

$$x_{1i}^T\left(\dot{M}_i(e_i) - 2C_i\left(e_i, \dot{e}_i\right)\right)x_{1i} = 0 \tag{47}$$

$$C_i(e_i, x_{1i})x_2 = C_i(e_i, x_{2i})x_{1i} \tag{48}$$

$$\left\|C_i(e_i, x_{1\,i})x_{2i}\right\| \leq \lambda_{ci}\|x_{1i}\|\|x_{2i}\|, \lambda_{ci} \geq 0 \tag{49}$$

$$C_i(e_i, x_{1i} + x_{2i})y_i = C_i(e_i, x_{1i})y_i + C_i(e_i, x_{2i})y_i \tag{50}$$

where $\lambda_{ci} \geq 0$ is a constant number.

**Theorem 2.** *By applying the bounded adaptive-robust controller in Equation (44), the formation errors in the leader-follower formation model of ASVs in Equation (26) converge into a small ball centered on the zero. Also. the estimated parameters in Equation (46) remain bounded.*

**Proof.** Substituting Equation (43), differentiating and increasing and decreasing the similar terms in Equation (39) results in Equation (51). □

$$\begin{aligned}
-\overline{\tau}_{ia}(t) &= M_i(e_i)\dot{r}_i + C_i\left(e_i, \dot{e}_i\right)r_i - M_i(e_i)\Lambda_{pi}\,sech^2(e_i)\,\dot{e}_i \\
&\quad -C_i\left(e_i, \dot{e}_i\right)\Lambda_{pi}\,tanh(e_i) + \xi_i
\end{aligned} \tag{51}$$

Now, by using Equations (48) and (50) and increasing and decreasing the similar terms in (51), the error dynamics in the closed-loop system is obtained as below:

$$\begin{aligned}
-\overline{\tau}_{ia}(t) &= M_i(e_i)\dot{r}_i + C_i(e_i, r_i)r_i - C_i(e_i, r_i)\Lambda_{pi}tanh(e_i) \\
&\quad -M_i(e_i)\Lambda_{pi}\,sech^2(e_i)\dot{e}_i - C_i\left(e_i, \Lambda_{pi}tanh(e_i)\right)r_i \\
&\quad +C_i\left(e_i, \Lambda_{pi}tanh(e_i)\right)\Lambda_{pi}\,tanh(e_i) + \xi_i
\end{aligned} \tag{52}$$

By substituting the controller from Equation (44), we have as follows:

$$M_i(e_i)\dot{r}_i = -C_i(e_i, r_i)r_i - K_{pi}tanh(e_i) - K_{iv}tanh(r_i) + u_{iR} - \xi_i + \chi_i \tag{53}$$

where:

$$\begin{aligned}
\chi_i &= M_i(e_i)\,\Lambda_{pi}Sech^2(e_i)\dot{e}_i - C_i\left(e_i, \Lambda_{pi}tanh(e_i)\right)\Lambda_{pi}\,tanh(e_i) \\
&\quad +C_i\left(e_i, \Lambda_{pi}tanh(e_i)\right)r_i + C_i(e_i, r_i)\Lambda_{pi}tanh(e_i)
\end{aligned} \tag{54}$$

Using property 2, $\chi_i$ will be bounded as follows:

$$\|\chi_i\| \leq \beta_1\|x_i\| + \beta_2\|x_i\|^2, \tag{55}$$

and $x_i = \left[tanh^T(e_i), r_i^T\right]^T$ where $\beta_1$ and $\beta_2$ are unknown positive constants.

To prove the stability of tracking error, the Lyapunov function defined as a function of the following variables:

$$V = \sum_{i=1}^{n} V_i\big(e_i, r_i, \widetilde{\theta}_i\big), i = 1, 2, \ldots, N \tag{56}$$

where for the *i*-th ASV, the Lyapunov function defined as below:

$$V_i = k_{pi} \ln \cosh(e_i) + \frac{1}{2} r_i^T M_i r_i + \frac{1}{2} \widetilde{\theta}_i^T \Gamma_\theta^{-1} \widetilde{\theta}_i \tag{57}$$

where $\widetilde{\theta}_i = \theta_i - \hat{\theta}_i$. Differentiating of (57) results in (58) and (59).

$$\dot{V}_i = tanh^T(e_i) K_{pi} \dot{e}_i + r_i^T M_i \dot{r}_i + 0.5 r_i^T \dot{M}_i r_i + \widetilde{\theta}_i^T \Gamma_\theta^{-1} \dot{\widetilde{\theta}}_i \tag{58}$$

By substituting Equations (43) and (53) into Equation (58), we have as follows:

$$\dot{V}_i = \frac{1}{2} r_i^T \big(\dot{M}_i - 2C_i\big) r_i - tanh^T(e_i) K_{pi} \Lambda_{pi} tanh(e_i)$$
$$- r_i^T K_{iv} tanh(r_i) + r_i^T u_{iR} - r_i^T \xi_i + r_i^T \chi_i + \widetilde{\theta}_i^T \Gamma_\theta^{-1} \dot{\widetilde{\theta}}_i \tag{59}$$

Using Equation (47), the first term in the derivative of the Lyapunov function equals zero. Thus, we have as follows:

$$\dot{V}_i = -tanh^T(e_i) K_{pi} \Lambda_{pi} tanh(e_i) - r_i^T K_{iv} tanh(r_i)$$
$$+ r_i^T u_{iR} - r_i^T \xi_i + r_i^T \chi_i + \widetilde{\theta}_i^T \Gamma_\theta^{-1} \dot{\widetilde{\theta}}_i \tag{60}$$

Now using (46) that is the adaptation rule and the relations $\xi_i \le G_i \theta_i$, $a^T b \le \|a\| \|b\|$ and $\lambda_{\min i}(M_i) \|x_i\|^2 \le x_i^T M_i x_i \le \lambda_{\max i}(M_i) \|x_i\|^2$, one can easily find a new upper bound for the first term of $\dot{V}_i$. Thus, we have as follows:

$$\dot{V}_i \le -\lambda_{min}\big\{K_{pi} \Lambda_{pi}\big\} \|tanh(e_i)\|^2 - \lambda_{di} \|r_i\|^2 - r_i^T K_{iv} tanh(r_i) + r_i^T u_{iR}$$
$$+ \|r_i^T\| G_i \theta_i + \|r_i^T\| \|\chi_i\| - \widetilde{\theta}_i^T G_i^T \|r_i\| + \widetilde{\theta}_i^T \delta_\theta\big(\hat{\theta}_i - \theta_0\big) \tag{61}$$

Using property $\|\chi_i\| \le \beta_1 \|x_i\| + \beta_2 \|x_i\|^2$ and assuming $-r_i^T K_{iv} tanh(r_i) \le 0$ and substituting $\|r_i^T\| G_i \hat{\theta}_i - r_i^T G_i \hat{\theta}_i tanh\big(\frac{G_i \hat{\theta}_i r_i}{\gamma_d}\big) \le n\gamma_{di}$ instead of $r_i^T u_{iR} - \|r_i^T\| G_i \theta_i$, according to [33] and property $ab \le \frac{(a^2 + b^2)}{2}$, we can easily write Equation (61) as below:

$$\dot{V}_i \le -\lambda_{min}\big\{K_{pi} \Lambda_{pi}\big\} \|tanh(e_i)\|^2 - (\lambda_{di} - 0.5\beta_1 - 0.5\beta_2) \|r_i\|^2$$
$$- c_{\theta_i} \big\|\widetilde{\theta}_i\big\|^2 + 0.5\beta_1 \|x_i\|^2 + 0.5\beta_2 \|x_i\|^4 + \gamma_i \tag{62}$$

where:

$$c_{\theta_i} = \Big(1 - \frac{0.5}{\kappa^2}\Big) \delta_{\theta_i} \tag{63}$$

$$\gamma_i = 0.5\delta_{\theta i} \kappa^2 \|\theta_i - \theta_0\|^2 + 2\gamma_{di} \tag{64}$$

According to $\lambda_{di} > 0.5\beta_1 + 0.5\beta_2$ and $\beta_m = \min\big\{\lambda_{min}\big\{K_{pi} \Lambda_{pi}\big\}, \{\lambda_{di} - 0.5\beta_1 + 0.5\beta_2\}\big\}$, we have as follows:

$$\dot{V}_i \le -\big(\beta_m - 0.5\beta_1 + 0.5\beta_2 \|x_i\|^2\big) \|x_i\|^2 - c_{\theta_i} \big\|\widetilde{\theta}_i\big\|^2 + \gamma_i \tag{65}$$

Now, assuming $\beta_m > 0.5\beta_1 + 0.5\beta_2 \|x_i\|^2$ it can be written as below:

$$c_m = \beta_m - 0.5\beta_1 + 0.5\beta_2 \|x_i\|^2 \tag{66}$$

$$\sum_{i=1}^{n} \dot{V}_i = \dot{V}_1 + \dot{V}_2 + \ldots + \dot{V}_n \le -\min((c_{m1}\|x_1\|^2 - c_{\theta_1}\|\widetilde{\theta}_1\|^2$$
$$+\gamma_1), \left(c_{m2}\|x_2\|^2 - c_{\theta_2}\|\widetilde{\theta}_2\|^2 + \gamma_2\right) + \ldots + \left(c_{mn}\|x_n\|^2 - c_{\theta_n}\|\widetilde{\theta}_n\|^2 + \gamma_n\right)) \tag{67}$$

As a result, $\dot{V}$ is strictly negative outside of the following set.

$$\Omega_x = \left\{ x_t(t) \big| 0 \le \|x_t(t)\| \le \sqrt{\gamma_i/c_m} \right\} \tag{68}$$

where $x_t = \left[x_i^T, \widetilde{\theta}_i^T\right]^T$. So, $\|x_t(t)\|$ remains bounded and considering saturation properties, it can be concluded that $e_i, r_i, \widetilde{\theta}_i \in L_\infty$.

## 4. Simulation Results

In this section, some examples have been simulated in MATLAB to show the effectiveness of the designed adaptive-robust controllers to keep leader-follower formation among ASVs.

### 4.1. Example 1

For the first example, consider a formation that consists of one leader and two followers. We apply the adaptive-robust control rule in Equation (32) and to avoid unboundedness in control inputs, a saturation function is applied on the control signals.

Consider the system dynamic matrices as below:

$$M_{1i} = \begin{bmatrix} m_{11i} & 0 \\ 0 & m_{33i} \end{bmatrix}, D_{1i} = \begin{bmatrix} d_{11i} & 0 \\ 0 & d_{33i} \end{bmatrix}$$
$$C_{1i}(v) = \begin{bmatrix} 0 & -m_{22i}v_i \\ (m_{22i} - m_{11i})v_i & 0 \end{bmatrix} \tag{69}$$

Proposed values of parameters for both followers and leader are shown in Table 1.

**Table 1.** ASV parameters.

| Parameter | Value |
|---|---|
| m11 | 25 (kg) |
| m22 | 25 (kg) |
| m33 | 2.5 (kg) |
| d11 | 7 (kg·m/s) |
| d22 | 7 (kg·m/s) |
| d33 | 5 (kg·m/s) |

The initial conditions of ASVs in the simulation are as follows:
Initial position of leader:

$$\eta_{ref} = [5, 10, \pi]^T \tag{70}$$

The initial position of followers:

$$\eta_1 = \left[-5, 10, \frac{\pi}{3}\right]^T, \eta_2 = \left[5, 1, \frac{\pi}{4}\right]^T \tag{71}$$

Desired values of relative distance and relative bearing for each follower:

$$q_{1d} = \left[4, \frac{\pi}{3}\right]^T, q_{2d} = \left[4, -\frac{\pi}{3}\right]^T \tag{72}$$

The control parameters for both follower ASVs:

$$
\begin{aligned}
&K_{1v} = \begin{bmatrix} 1 & 0 \\ 0 & 2 \end{bmatrix}, K_{2v} = \begin{bmatrix} 1 & 0 \\ 0 & 2 \end{bmatrix}, K_{\varepsilon 1} = 1, \; K_{\varepsilon 2} = 1, \gamma_1 = 4, \gamma_2 = 4 \\
&\varepsilon_1(0) = 1, \varepsilon_2(0) = 1 \\
&\hat{\theta}_1(0) = [1, 2, 3, 2, 3, 1]^T \\
&\hat{\theta}_2(0) = [1, 2, 3, 2, 3, 1]^T
\end{aligned}
\tag{73}
$$

It is also assumed that the input torque, i.e., $\tau_{ai} = [\tau_u, \tau_r]^T$ for both follower ASVs are constrained by a saturation function with saturation levels of $\pm 120$. Simulation results are shown in Figures 5–8.

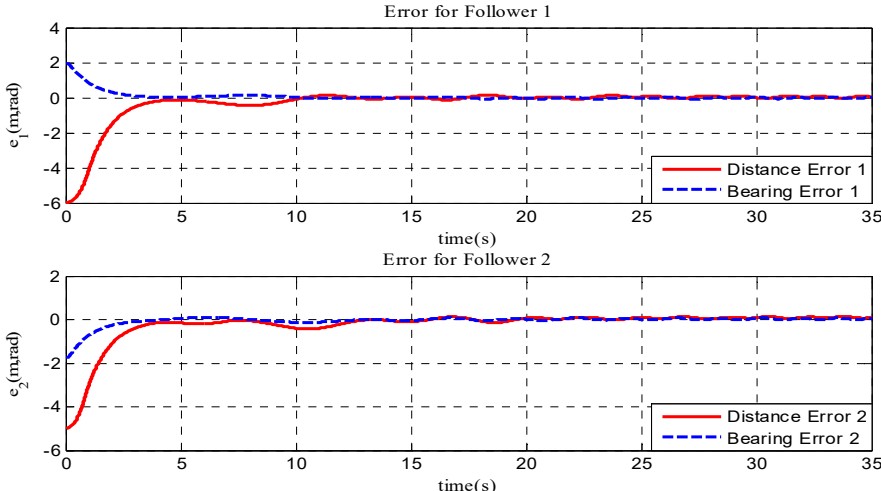

**Figure 5.** Relative distance and relative bearing errors of followers in Example 1.

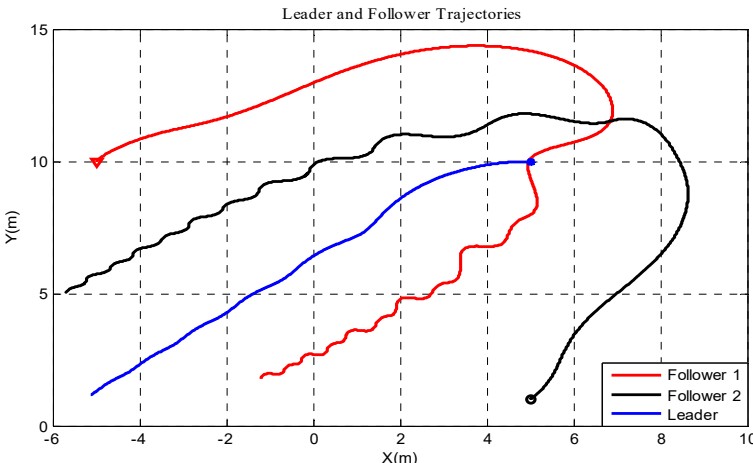

**Figure 6.** Leader-follower trajectories in Example 1.

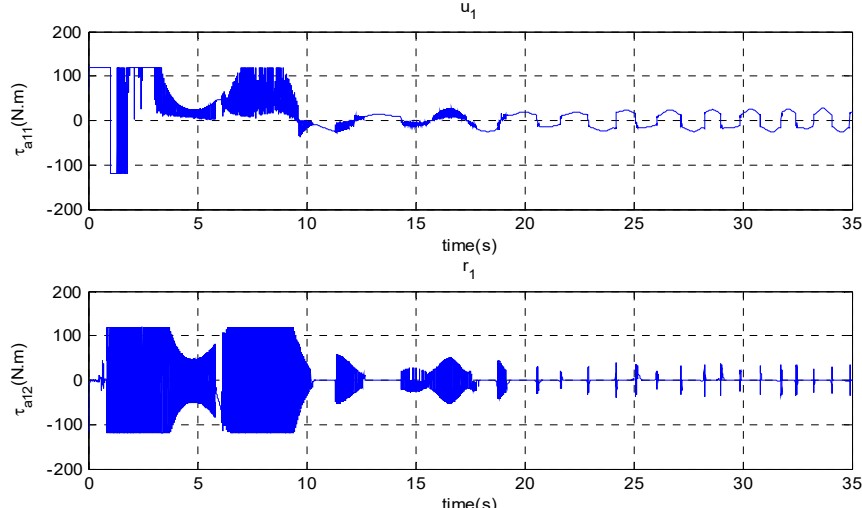

**Figure 7.** Applied input controls of the first follower in Example 1.

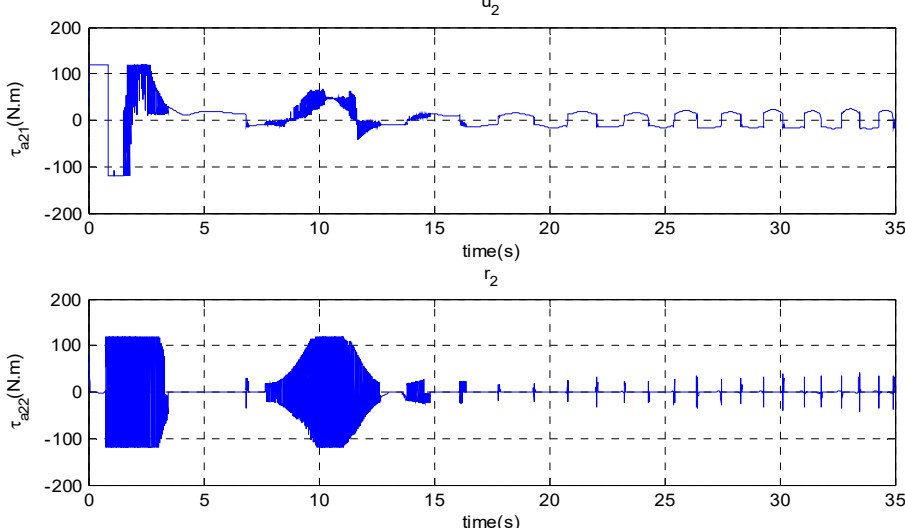

**Figure 8.** Applied input control of the second follower in Example 1.

It is clear from Figure 5 that relative distance $\rho_i(t)$ and relative bearing $\psi_i(t)$ errors for both followers go to zero. This means that the controller has been successful to keep formation. Moreover, the trajectories of follower ASVs and leader ASV in Figure 6 properly shows the desired leader-follower formation starting from the stated initial conditions of ASVs.

Input torques for each of follower vessels are shown in Figures 7 and 8. As shown in these figures, the oscillations in the input controls, especially before reaching the desired formation are too much and this makes it difficult to be applied in practice. This volatility is because of the discontinuity of the saturation function. As shown in the second example, using a hyperbolic tangent function as a continuous function, we can overcome this chattering.

Figure 9 shows the estimated vector $\hat{\theta}_i = [\hat{\delta}_0, \hat{\delta}_1, \hat{\delta}_2]^T, i = 1, 2$ parameters for each follower in the leader-follower formation.

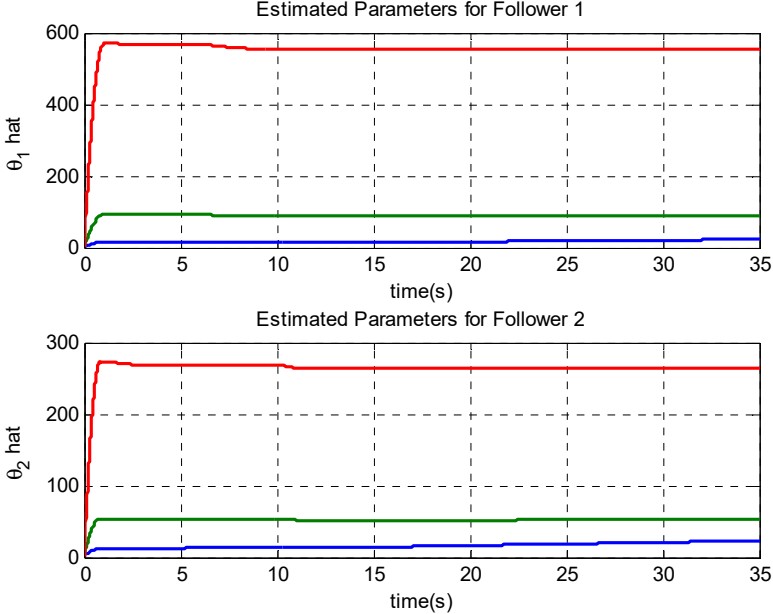

**Figure 9.** Estimated value of the vector $\hat{\theta}_i$.

### 4.2. Example 2

For the second Example, considering four followers and one leader in a formation, the control rule in Equation (44) has been simulated. In this case using a hyperbolic tangent function that is a bounded function, we expect that input controllers remain bounded. In Example 1, due to irregularity and the discontinuity of saturation function, the oscillations in the input torques are severe. However, in the second simulation, due to the hyperbolic tangent function that is smooth and continuous, in addition to the constraint of the control signals, the discontinuities in the input controls intensity decreased.

Desired values of relative distance and relative bearing for each follower are as follows:

$$q_{1d} = \left[3.5, \frac{\pi}{3}\right]^T, q_{2d} = \left[3.5, -\frac{\pi}{3}\right]^T$$
$$q_{3d} = \left[1.5, \frac{\pi}{3}\right]^T, q_{4d} = \left[1.5, -\frac{\pi}{3}\right]^T \tag{74}$$

The control parameters for all of followers are as below:

$$K_{pi} = \begin{bmatrix} 25 & 0 \\ 0 & 25 \end{bmatrix}; K_{iv} = \begin{bmatrix} 10 & 0 \\ 0 & 10 \end{bmatrix};$$
$$\delta_{\theta_i} = 0.0001, i = 1, \ldots, 4 \tag{75}$$

$$\Lambda_{pi} = \begin{bmatrix} 0.25 & 0 \\ 0 & 2.5 \end{bmatrix}, \gamma_{di} = 1 \tag{76}$$

$$\Gamma_{\theta_i} = \mathrm{diag}(10, 10, 10, 0.1), i = 1, \ldots, 4 \tag{77}$$

The initial value of parameters in estimation is considered as below:

$$\hat{\theta}_i = [1, 3, 2, 1]^T, i = 1, 2, \ldots 4 \tag{78}$$

In this case, simulation results are shown in the Figures 10–15.

As it is shown in Figure 10, relative errors for all four followers goes to zero; this confirms the effectiveness of the controller for keeping leader-follower formation. Although a steady-state error can be seen at the end of the convergence. The trajectories of ASVs in the formation are shown in Figure 11 that confirms leader-follower formation.

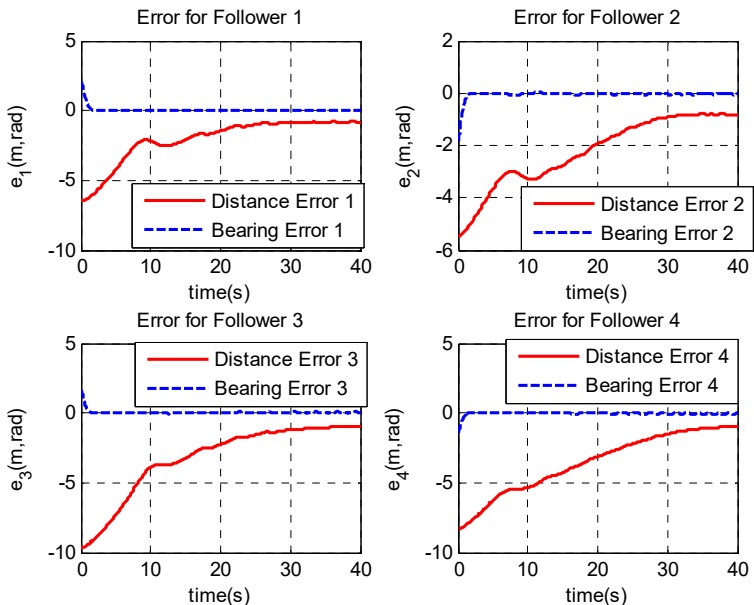

**Figure 10.** Relative distance and relative bearing errors of followers in Example 2.

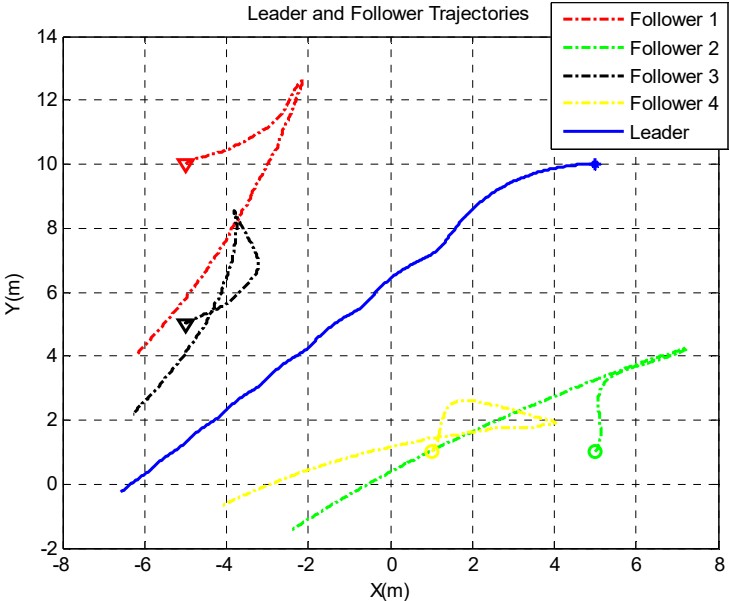

**Figure 11.** Leader-follower trajectories in Example 2.

In this case, we expect that discontinuity in the control input is less than as for the first simulation. To illustrate this, the control input of the first follower is shown in Figure 11. The oscillation and discontinuity in much less in comparison with previous simulations where the only function saturation was used. Indeed, function tanh that is used in the controller (44), unlike the saturation function, is a continuous function and as expected, acted well in reducing the amount of discontinuity and chattering. Torque inputs are shown for each of the four follower vessels in Figures 12–15.

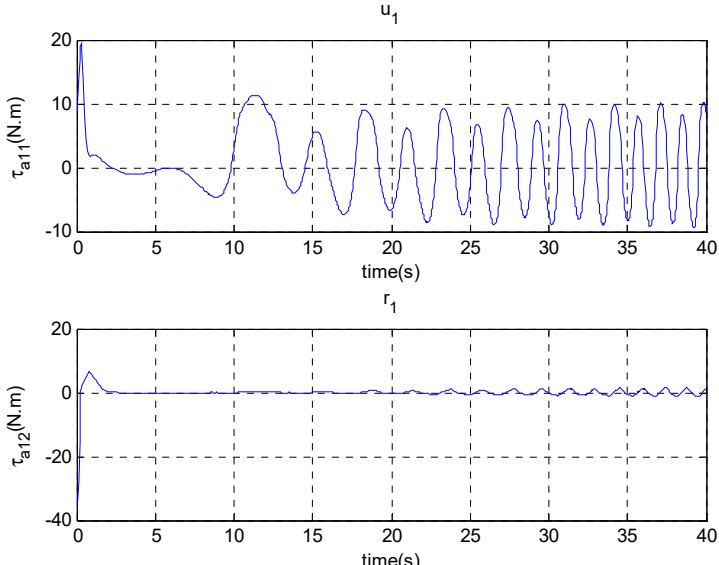

**Figure 12.** Applied input control of the first follower in Example 2.

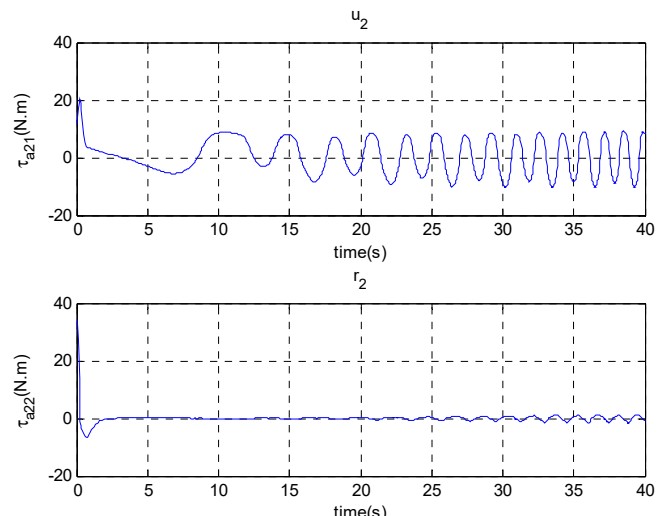

**Figure 13.** Applied input control of the second follower in Example 2.

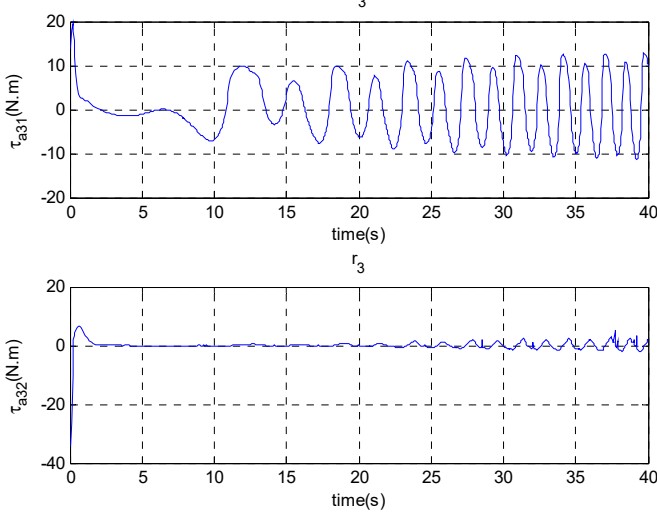

**Figure 14.** Applied input control of the third follower in Example 2.

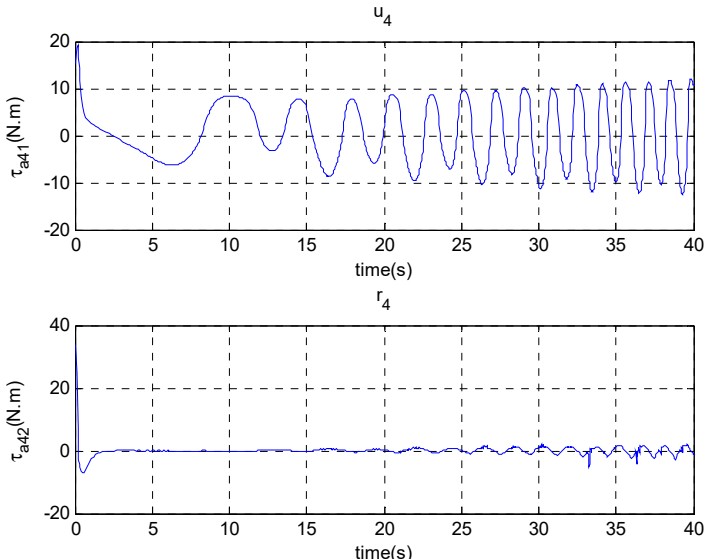

**Figure 15.** Applied input control of the fourth follower in Example 2.

As seen from these figures, the volatility and discontinuity were much lower than the simulation of the previous section, where only one saturation function was used. Indeed, the hyperbolic tangent function used in contrast to the controller (44) is in contrast to the functional saturation function and, as we have anticipated, worked well to reduce the amount of discontinuity or chattering. In Figure 16, the hyperpolecular tangent function $(\tanh(e_i))$ and $(\ln(\cosh e_i))$ function are plotted in terms of a function of error.

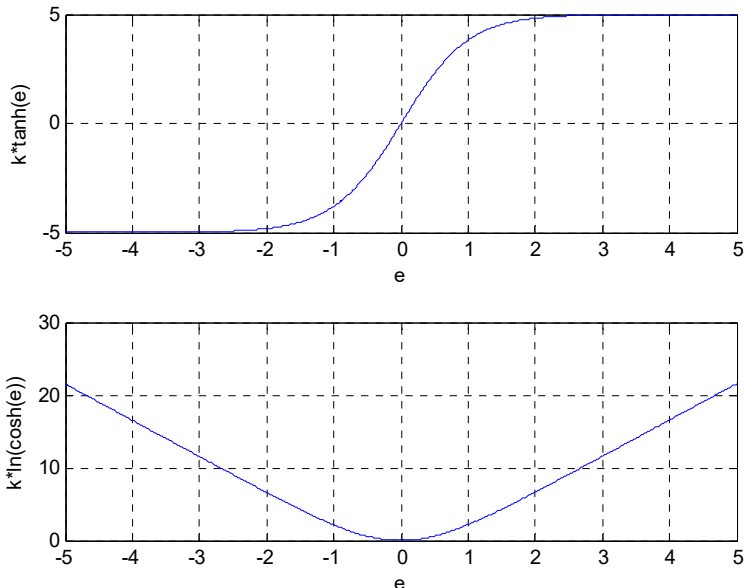

**Figure 16.** Hyperbolic tangent function and its derivative in error $e_i$.

Figure 17 shows the estimated vector $\hat{\alpha}_i = [\hat{\delta}_0, \hat{\delta}_1, \hat{\delta}_2]^T, i = 1, 2, \ldots 4$ parameters for each follower in the leader-follower formation.

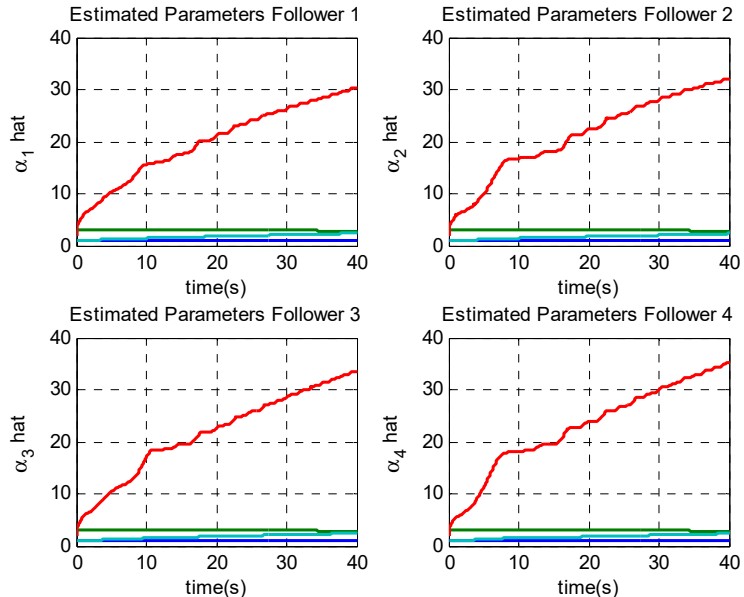

**Figure 17.** Estimated value of the vector $\hat{\alpha}_i$.

## 5. Conclusions

In this paper, we studied leader-follower formation control of underactuated autonomous surface vehicles (ASVs) in the presence of model uncertainties and input saturation constraints. First, an adaptive-robust controller designed to keep formation in the presence of model uncertainties and external disturbances. Moreover, a saturation function is added to this controller to bound input torques of the followers. Although this function limits the input signals, it also causes chattering and oscillation. Then, a new adaptive robust controller designed to keep formation in the presence of input saturation constraints in addition of model uncertainties. This controller includes a hyperbolic tangent function that results in a smooth bounded controller. Based on the Lyapunov synthesis, it is proven that by applying this controller, the closed-loop system is stable and all the formation errors converge to a small neighborhood of zero. Some simulation results presented to illustrate the operation of the proposed controllers.

**Author Contributions:** All the authors conceived the idea, developed the method, and conducted the experiment. A.R. contributed to the formulation of methodology and experiments. S.M.H.R. contributed in the data analysis and performance analysis and contributed to the overview of the proposed approach and decision analysis. J.W. and H.J.K contributed to the algorithm design and data sources. All authors read and approved the final manuscript.

**Funding:** This work is funded by the National Natural Science Foundation of China (grant number 61772454, 61811530332, 61811540410).

**Conflicts of Interest:** The authors declare no conflict of interest.

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
