# Peer review of "Adaptive Leader-Follower Formation Control of Under-actuated Surface Vessels with Model Uncertainties and Input Constraints"

_applsci, doi:10.3390/app9183901_

Round 1
Reviewer 1 Report
The paper is very interesting and shows actual problems faced in unmanned vehicles community. Especially in the filed of transportation safety. The algorithm, as showed in the research is working and output is satisfactory for presented thesis. Form the practical point of view, I would generate question to be answered by the author, if of course they see the practical usage of this research.
On the typical USV in formation, where the presented algorithm will be implemented, centrally of on each USV separately ?Form the research I understood that simulation was done centrally, means that this algorithm controlled the group of USV. So in pratical usage it should be implemented on the leader and relative position of flowers should be passed ?
Author Response
The paper is very interesting and shows actual problems faced in unmanned vehicles community. Especially in the filed of transportation safety. The algorithm, as showed in the research is working and output is satisfactory for presented thesis. Form the practical point of view, I would generate question to be answered by the author, if of course they see the practical usage of this research.
1. On the typical USV in formation, where the presented algorithm will be implemented, centrally of on each USV separately ?
Author reply: A surface vessel is designated as a leader and is responsible for directing the overall arrangement.
Other surface vessels (followers) should be controlled in such a way as to follow the leader's surface vessel at predetermined intervals. Adaptive controllers are used to maintaining the leader-follower arrangement.
2. Form the research I understood that simulation was done centrally, means that this algorithm controlled the group of USV. So in pratical usage it should be implemented on the leader and relative position of flowers should be passed ?
Author reply: Yes. It is true.
Reviewer 2 Report
- There are still some grammatical errors, even in introduction section that must be fixed.
- All the figures and tables should be moved to the top of page.
- There is not any individual section for background/related work. It has been mentioned as introduction, while the introduction part should briefly introduce the problem, motivation of the work, aims and research ideas and finally the contribution of work very clearly e.g. in bullet points. I encourage the authors to have two different sections, i.e. introduction and related work, while cutting down and moving the literature review stuff to the section called Related work; and adding new introduction stuff to the introduction section. Also, literature review has been written in a format of mapping whatever other researchers done before. Better to mention the gaps in the research area that have not been addressed yet and their correlation with this paper.
- The representation of formulas is not consistent along the paper, sometimes cantered sometimes in the left-hand side. For instance, equations 2-6 versus 7-10.
Author Response
- There are still some grammatical errors, even in introduction section that must be fixed.
We edited style of sections and also language.
- All the figures and tables should be moved to the top of page.
Text format is such that not all figures and tables can be moved exactly to the top of the page.
- There is not any individual section for background/related work. It has been mentioned as introduction, while the introduction part should briefly introduce the problem, motivation of the work, aims and research ideas and finally the contribution of work very clearly e.g. in bullet points. I encourage the authors to have two different sections, i.e. introduction and related work, while cutting down and moving the literature review stuff to the section called Related work; and adding new introduction stuff to the introduction section. Also, literature review has been written in a format of mapping whatever other researchers done before. Better to mention the gaps in the research area that have not been addressed yet and their correlation with this paper.
All of the items you mentioned were applied to the text and highlighted.
- The representation of formulas is not consistent along the paper, sometimes cantered sometimes in the left-hand side. For instance, equations 2-6 versus 7-10.
We corrected it.
Reviewer 3 Report
Although the response of the authors is different from my comment and intention, there may be no need for another round of review. The organization of the manuscript is anyway better.
Author Response
Although the response of the authors is different from my comment and intention, there may be no need for another round of review. The organization of the manuscript is anyway better.
Ok. Thanks.
Round 2
Reviewer 2 Report
Presentaion and writing of the munuscript has been improved.
You can still improve it in your revised version.
This manuscript is a resubmission of an earlier submission. The following is a list of the peer review reports and author responses from that submission.
Round 1
Reviewer 1 Report
The paper presents Adaptive Leader-Follower Formation Control of 3 Under-actuated Surface Vessels with Model 4 Uncertainties and Input Constraints.
The iterature reviev is extensive and covers the topic. The method is described in details and results are presented clearly.
There are some spelling and typing mistakes, to be corected.
Overal paper nad method is presented with high scientific discipline.
The topic is also becoming very demanding by egeniers working on this problems and become in shot term very damanding by unmaned industry.
It would be perfect if reserchers would share matlab code with other reserachers and enginiers, nevertheles the method was presented in the paper.
Author Response
Reviewer #1:
The paper presents Adaptive Leader-Follower Formation Control of 3 Under-actuated Surface Vessels with Model 4 Uncertainties and Input Constraints.
The literature review is extensive and covers the topic. The method is described in details and results are presented clearly.
There are some spelling and typing mistakes, to be corrected.
Done.
Overal paper nad method is presented with high scientific discipline.
The topic is also becoming very demanding by engineers working on these problems and become in a shot term very demanding by unmanned industry.
It would be perfect if researchers would share Matlab code with other researchers and engineers, nevertheless, the method was presented in the paper.
Ok.

Reviewer 2 Report
Summary:
In this paper, an adaptive-robust controller is proposed to keep formation in the presence of model uncertainties and external disturbances. Another adaptive robust controller is also presented for formation control of ASV in the presence of input saturation constraints and model uncertainties.
Strong points:
- The description of design part is understandable. They considered external disturbances, model uncertainties and input saturation constraints.
- Simulation results are enough to support the claim but must be presented in a better way. (There are lots of issues in presentation of figures that I will mention some in the following paragraphs.)
Weak points:
- The paper’s writing needs to be polished. I suggest authors to revise it professionally. There are grammatical errors, typos, punctuations issues, extra spaces between words, and misuse of terms, e.g. pre-described trajectory should be replaced by pre-defined trajectory. Another example: they used this ill-formed style a lot: This [29] refers to the issue of …
- Conclusion and abstract must be well-written and are not acceptable in the current format.
- In the introduction section, when explaining the motivation of work and applications, references are better to be mentioned right after each application as the following:
Relevant applications include automatic ocean exploration [ref], environmental monitoring [ref], disaster search and rescue [ref], …
- Each related work should be briefly explained, unless there is a close resemblance between your work and theirs, which you would need to explain more to clarify differences.
- References for similar adaptive controllers that have been used previously in the marine device’s literature are not enough. If there are not enough references in ASV you can refer to adaptive trajectory tracking control of an AUV, in which the degree of freedom is higher but theory of controller is the same. Good examples:
1.A novel approach to 6-DOF adaptive trajectory tracking control of an AUV in the presence of parameter uncertainties.
2.Design of an adaptive nonlinear controller for an autonomous underwater vehicle.
- The type of stability was not mentioned neither in design part nor in conclusion, please specify you obtained the asymptotical or exponential stability. Also, more technical terms are required when you’re summarising your achievements in a short conclusion.
- Figures presences should be improved. E.g. in Figure 10, authors used red colour for two different followers and black colour for other two followers which does not make any sense. They could simply use a unique colour for each follower.
- In Figure 9, legend box covers one of the results indicated by green colour, which is inappropriate.
- Formulations writing also needs to be fixed. (font is large and inconsistent)
- The material which was provided for property.1 seems to be need more explanation. It’s just mentioned that it can be proven, but based on what?
- In the following paragraph, readers were referred to a previous work for checking the stability, which is not acceptable. It was better to have the stability proof in appendix, if you have the lack of space issue:
- Proof: The proof is similar to the proof of applying a similar controller that has been proposed in [28] for solving the desired path following problem in manipulators. To prevent recurrence, the proof is omitted here.
Author Response
Reviewer #2:
Summary:
In this paper, an adaptive-robust controller is proposed to keep formation in the presence of model uncertainties and external disturbances. Another adaptive robust controller is also presented for formation control of ASV in the presence of input saturation constraints and model uncertainties.
Strong points:
- The description of design part is understandable. They considered external disturbances, model uncertainties and input saturation constraints.
- Simulation results are enough to support the claim but must be presented in a better way. (There are lots of issues in presentation of figures that I will mention some in the following paragraphs.)
Weak points:
- The paper’s writing needs to be polished. I suggest authors to revise it professionally. There are grammatical errors, typos, punctuations issues, extra spaces between words, and misuse of terms, e.g. pre-described trajectory should be replaced by pre-defined trajectory. Another example: they used this ill-formed style a lot: This [29] refers to the issue of …
- Conclusion and abstract must be well-written and are not acceptable in the current format.
Done.
- In the introduction section, when explaining the motivation of work and applications, references are better to be mentioned right after each application as the following:
Relevant applications include automatic ocean exploration [ref], environmental monitoring [ref], disaster search and rescue [ref], …
Done.
- Each related work should be briefly explained, unless there is a close resemblance between your work and theirs, which you would need to explain more to clarify differences.
Done.
- References for similar adaptive controllers that have been used previously in the marine device’s literature are not enough. If there are not enough references in ASV you can refer to adaptive trajectory tracking control of an AUV, in which the degree of freedom is higher but theory of controller is the same. Good examples:
1.A novel approach to 6-DOF adaptive trajectory tracking control of an AUV in the presence of parameter uncertainties.
2.Design of an adaptive nonlinear controller for an autonomous underwater vehicle.
Done.
- The type of stability was not mentioned neither in design part nor in conclusion, please specify you obtained the asymptotical or exponential stability. Also, more technical terms are required when you’re summarising your achievements in a short conclusion.
Done.
- Figures presences should be improved. E.g. in Figure 10, authors used red colour for two different followers and black colour for other two followers which does not make any sense. They could simply use a unique colour for each follower.
We use different colors for that the reader understands good the problem.
- In Figure 9, legend box covers one of the results indicated by green colour, which is inappropriate.
61/5000
To better understand the subject, we do this so that the reader is not confused.
- Formulations writing also needs to be fixed. (font is large and inconsistent)
Done.
- The material which was provided for property.1 seems to be need more explanation. It’s just mentioned that it can be proven, but based on what?
Done.
- In the following paragraph, readers were referred to a previous work for checking the stability, which is not acceptable. It was better to have the stability proof in appendix, if you have the lack of space issue:
- Proof: The proof is similar to the proof of applying a similar controller that has been proposed in [28] for solving the desired path following problem in manipulators. To prevent recurrence, the proof is omitted here.
Done.
Reviewer 3 Report
I think this manuscript needs a thorough proofreading. I gave up a deeper reading even in the introduction. This manuscript is basically not organized by the IMRAD structure, apart from the poor English writing. In the five-page introduction, what paragraph does correspond to the state of the art? Where is a short explanation for the paper organization located? It is really confusing.
Moreover, as the technical comments, the strength of this manuscript seems to be weak. In the application of the control theory, it is important that the assumption of the study is realistic. Otherwise, there are too many occasions for the combination of the leader and follower formation and most of them are meaningless. The reviewer understands what the authors have studied, but doesn't know why the authors did this work. Of course, it is also due to the poor quality of the writing.
Author Response
Reviewer #3:
I think this manuscript needs a thorough proofreading. I gave up a deeper reading even in the introduction. This manuscript is basically not organized by the IMRAD structure, apart from the poor English writing. In the five-page introduction, what paragraph does correspond to the state of the art? Where is a short explanation for the paper organization located? It is really confusing.
We improved the English text. In Introduction, We in first two paragraphs gave an explanation of what the work was going to be. We then pay to the background in the next paragraphs. Designing a control rule in such a way that the surface vessel begin to move from an arbitrary state and in that case becomes favorable to the stability. The surface vessel needs to follow a parameterized time reference. For surface vessels with full stimulation, the problem can be solved with advanced nonlinear control rules. Both the position of the center of mass of the vessel point position are much more important than the third degree of freedom, ie, the angle φ. In the case of incomplete stimulation, it is a subject of interest for research.
Moreover, as the technical comments, the strength of this manuscript seems to be weak. In the application of the control theory, it is important that the assumption of the study is realistic. Otherwise, there are too many occasions for the combination of the leader and follower formation and most of them are meaningless. The reviewer understands what the authors have studied, but doesn't know why the authors did this work. Of course, it is also due to the poor quality of the writing.
We improved the English text. There are a lot of things in the real world that is used of this method Follower-Leader. For example: marine-ocean explorations, naval-related military environments, use in unknown and dangerous environments, the displacement of large or special platforms or vessels in the oceans or seas, identification, monitoring and surveillance in offshore and offshore waters. Collect environmental information on board ships
Round 2
Reviewer 3 Report
Add the specific and realistic applications of the method (Leader‐follower formation control of underactuated autonomous surface vehicles in the presence of model uncertainties and input saturation constraints ) in the manuscript, not the general description of the leader-follower control.
Author Response
Add the specific and realistic applications of the method (Leader‐follower formation control of underactuated autonomous surface vehicles in the presence of model uncertainties and input saturation constraints ) in the manuscript, not the general description of the leader-follower control.
Done. On pages 5,6 and 7, we noted that we did this for Marine floating.